# Sectoral linkages and their influence on structural change: The case of China

Tao Jin, Zhihao Li 🔾 *

School of Economics, Xiamen University, Xiamen, Fujian Province, China

* andycristone@163.com

## Abstract

Structural change is a fundamental aspect of a country's economic development process. Unlike traditional literature, our research integrates input-output analysis with a multisector general equilibrium model to investigate the role of sectoral input-output linkages throughout this process. By utilizing this framework and applying it to Chinese data, we find that sectoral input-output linkages are as critical as final demand in our analysis. The stability of the manufacturing sector and the prosperity of the services sector are significantly influenced by these linkages. Our findings remain robust in counterfactual analyses, even after modifying the model specifications. We apply methodologies from production networks research to elucidate the mechanisms by which linkages function. Through structural decomposition analysis, we identify the various components that constitute these driving forces. Ultimately, our results align closely with our baseline findings.

## Introduction

Structural change or transformation, is recognized as a fundamental characteristic of economic development. This process can be divided into two phases: the transformation from an agriculture-dominated economy to one dominated by manufacturing, followed by a subsequent shift from manufacturing to one that is service-oriented. Throughout this whole transformation process, there is a decline in the agricultural sector's share, a hump-shaped path in manufacturing, and an increase in the share of services measured by value-added or employment shares within the economic development. This phenomenon is commonly referred to as the "Kuznets Facts." Since the implementation of reforms and the opening up of its economy in 1978, China has experienced remarkable growth, often described as an economic miracle, which has entailed significant structural transformation. China has effectively completed its first phase of this transformation, transforming from an agriculture-dominated economy to one that has developed a relatively comprehensive industrial system.

**Data availability statement:** All relevant data are within the paper and its Supporting Information files.

**Funding:** The author(s) received no specific funding for this work.

**Competing interests:** The authors have declared that no competing interests exist.

To study the structural transformation process in China, we utilized the data from the World Input-Output Database (WIOD) relevant to China from 1978 to 2014. The WIOD provides input-output data for sectoral production and final demand, which is essential for examining the function of sectoral linkages. We adopt the standard framework of Herrendorf et al. (2014) [1] to aggregate subsectors in WIOD into three broad sectors: agriculture, manufacturing, and services. Subsequently, we depict the dynamics of their value-added shares and sectoral linkages.

Fig 1 illustrates the dynamics of sectoral value-added shares in China from 1978 to 2014. It depicts the typical decline in the share of agricultural sector and the increase in the services sector; a phenomenon commonly observed in the structural transformation of most countries. However, a pronounced hump-shaped pattern in the manufacturing sector is not apparent; rather, its share exhibits slight fluctuations and remains more stable compared to the other two sectors. Notably, between 2013 and 2014, the share of services surpassed that of manufacturing. While the value-added share of services has exceeded that of the manufacturing sector in 2013, the latter continues to retain a certain proportion within the economy.

Fig 2 illustrates the dynamics of sectoral intermediate inputs across three sectors, depicted from left to right. In the agricultural sector, the input from manufacturing has steadily increased since 2004. Conversely, intra-sector input has declined, while the input from services has experienced a modest increase. In the manufacturing sector, intra-sector input remains at a high level, exhibiting only minor fluctuations. Agricultural input has also shown a gradual decrease, whereas services input has remained relatively stable. In the services sector, there exists a noticeable upward trend in intra-sector input. Although the share of manufacturing input has decreased, it still accounts for over 40% in 2014, indicating its significant contribution. Agricultural input remains low with little variation. Thus, it can be observed that there are common characteristics among the shares of sectoral intermediate inputs and value-added in China. A notable feature is the role of the manufacturing sector, which reflects its influence on value-added shares. This figure illustrates that the manufacturing sector exhibits a higher degree of interconnection in production compared to the other two sectors.

What insights can be derived from these observations? It can be posited that China demonstrates distinct characteristics in its structural transformation. The figures above illustrate that although the services sector is increasingly influential within the economy, manufacturing continues to maintain a critical role.

In this context, we attempt to elucidate the impacts of sectoral linkages on the Chinese structural transformation process. Specifically, we aim to explore how these linkages function throughout this process and through which mechanisms they operate. Furthermore, we investigate how the distinct structures within the three broad sectors contribute to the dynamics of sectoral growth. By addressing these inquiries, our study tries to analyze the impacts of sectoral linkages from a long-term perspective on sectoral growth.

Our study enriches the existing body of literature concerning structural change in China from the view of intermediate goods production. Leif [2] has concluded four

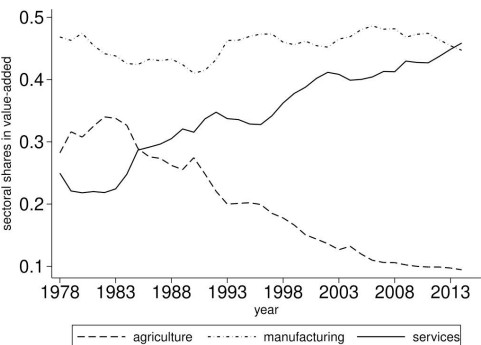

**Fig 1. Sectoral value-added shares in China from 1978 to 2014.** The horizontal axis denotes the years and the vertical axis denotes the share of value-added attributed to each sector.

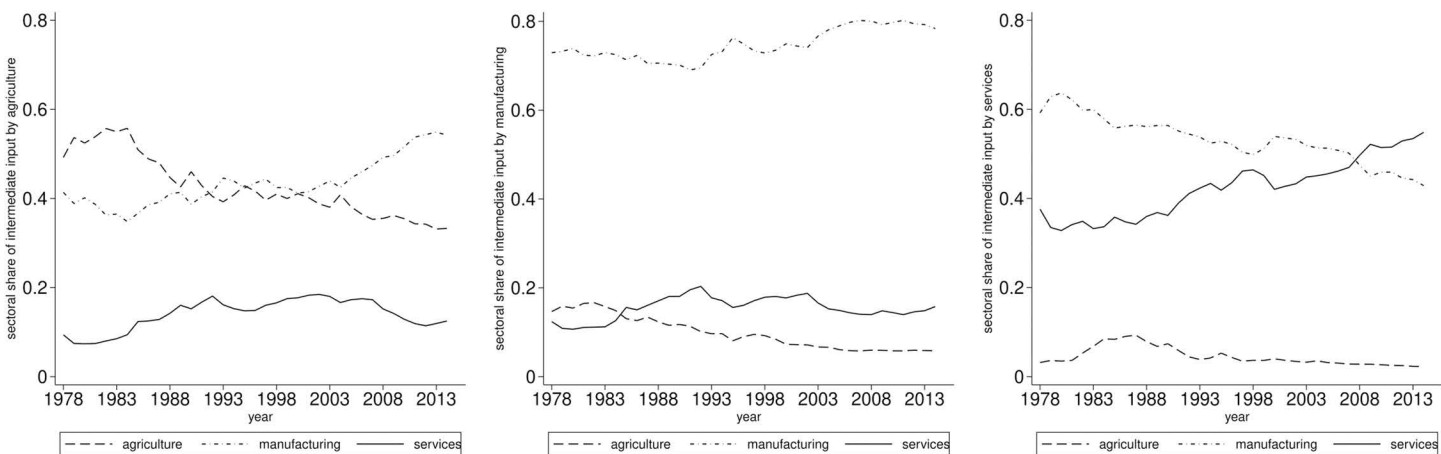

**Fig 2. Sectoral shares of intermediate inputs in China from 1978 to 2014.** The panels arranged from left to right illustrate the annual variations in sectoral shares of intermediate inputs across agriculture, manufacturing, and services.

driving forces behind the structural transformation process: (1) the income effects stemming from final demand for various sectoral goods, (2) changes in the relative prices of sectoral goods driven by productivity, (3) changes in sectoral input-output linkages concerning intermediate inputs, and (4) changes in a sector's comparative advantage due to international trade. A substantial amount of research has thoroughly investigated the demand-driven and supply-driven factors represented by (1) and (2), as well as the impacts of trade as outlined in (4). Compared to these factors, the examination of sectoral linkages as a driving force behind structural change remains relatively underexplored, with most scholars focusing on their impacts on economic fluctuations and aggregate volatility from a production network perspective. Unlike studies that mainly focus on final goodsdemand or technological advancement, our study highlights the significance of sectoral linkages in production process, as indicated by the input-output structure. In doing so, it complements the current literature by emphasizing that the utilization of intermediate goods in production and their dynamic variations is as essential as the demand for final goods and technological advancements in understanding sectoral value-added dynamics. We examine the functions of sectoral linkages from a production standpoint and their relationship with final output and consumption.

Additionally, this study contributes to the literature on production networks and economic development. Traditional literature has provided substantial evidence that horizontal fragmentation among sectors significantly impacts the structural

transformation process from a demand-driven perspective. In contrast, we present evidence supporting the role of vertical fragmentation from the view of intermediate inputs. This highlights the long-term effects of productivity growth propagating through production networks, particularly the input-output structure, on structural change within an economy as demonstrated through cross-country comparisons. The findings confirm that production networks not only affect aggregate volatility and economic fluctuations in the short term but also influence sectoral value-added dynamics in the long term.

Finally, this study contributes to the field of input-output analysis by integrating structural decomposition analysis (SDA) within a general equilibrium framework. We establish a connection of different components in sectoral value-added shares under the SDA framework with their corresponding elements in the general equilibrium model. This integration provides a theoretical foundation and microeconomic mechanism for the implication of SDA, thereby enhancing its application in research.

The remainder of this paper is structured as follows: The *Literature review* section reviews the relevant literature. The *Methodology* section outlines the methodologies we use in our research. The *Quantitative results* section presents corresponding numerical results of the methodologies discussed in the *Methodology* section. The *Conclusion* section concludes our research.

## Literature review

Sectoral linkages, defined as the input-output relationships among various sectors, are essential to understanding structural change. This section primarily reviews relevant literature and its contributions to the field. Our research is guided by three distinct strands of literature.

The first relevant body of literature examines the economic effects of variations in sectoral linkages. Several studies have focused on fluctuations in the demand for intermediate inputs within the sectoral production process. Duarte and Restuccia [3] argue that it is crucial to incorporate sectoral linkages into the calculation of a sector's TFP based on the prices of final goods. Meanwhile, Fadinger et al. [4] assert that sectoral linkages contribute to aggregate productivity. Their findings indicate that low TFP sectors are more interconnected in wealthier countries compared to poorer ones, which help reduce income disparities within those societies. Bartenlme and Goronichenko [5] find that the presence of distortions leads to differences in the level of sectoral linkages. This situation diminishes income in poorer countries, as they tend to experience more pronounced distortions compared to their wealthier counterparts.

Berlingieri [6] employs a partial equilibrium analytical framework to illustrate that the increasing reliance on relevant business and professional services as intermediate inputs in the United States significantly affects the employment dynamics within the services sector. Additionally, Sposi [7] presents international evidence regarding the dynamics of intermediate demand within a general equilibrium framework. He argues that sectoral linkages exhibit distinct patterns relating to income per capita across different countries. His research reveals that agricultural production in wealthier countries utilizes intermediates inputs more intensively than in poorer countries. Furthermore, wealthier countries employ services more intensively than their poorer counterparts across all sectors, highlighting the fact that wealthier countries have become service-dominant in their economic structures.

The second relevant body of literature is about production networks and their significance in economic development. Conventional macroeconomic models often omit intermediate inputs in the production function. To investigate the role of these inputs in production, some researchers have begun to incorporate them into production functions. Production functions that incorporate an input-output structure can effectively depict sectoral linkages in the framework of a production network, which significantly affects economic development from multiple perspectives. Carvalho and Tahbaz-Salehi [8] offer a comprehensive review of production networks within the context of macroeconomics, establishing a theoretical foundation of the sectoral linkages as channels for propagating shocks and transforming these shocks into macroeconomic fluctuations. Their work encompasses both empirical and theoretical literature on this subject. The multisector growth model that incorporates sectoral linkages can be traced back to the contributions of Long Jr and Plosser [9]. They

find that an exogenous shock to one sector can propagate through linkages, resulting in economic fluctuations at the aggregate level. This is known as real business cycle theory. Acemoglu et al. [10] further investigate the upstream propagation of demand-side shocks and the downstream propagation of supply-side shocks. They decompose these shocks into the direct and indirect effects, which can be observed within the production network. Jones [11] develops a theoretical model that incorporates intermediate goods to examine how the multipliers associated with these goods influence misallocation in the production process. They also assess the impacts of these factors on the cross-country income gap. Additionally, Baqaee and Farhi [12] address distortion and misallocation in an inefficient economy, concluding that eliminating misallocation and improving allocative efficiency can promote TFP growth by reallocating market share to firms with high markup, with their model accommodating input-output network linkages. In contrast to a predetermined production network, Oberfield [13] proposes a theory that makes firms' decision on purchasing intermediate goods endogenous. Based on this model, they explore the structure of the input-output network and its impacts on firms' productivity within an economy. Furthermore, Acemoglu and Azar [14] develop an endogenous production network to analyze the processes of productivity generation and distortion, revealing that technological advancements spread throughout the economy via input-output linkages. The endogenous evolution of the production network may serve as a significant driving force for sustained economic growth.

The last relevant body of literature is about input-output analysis. The input-output analysis is introduced by W. Leontief, who constructs a balanced table to reflect the flows of goods among various sectors in terms of production and consumption. Furthermore, he also introduces several statistics to illustrate sectoral linkages. Since then, his idea and work have inspired many scholars to research in this domain. The table he uses to analyze the economic structure is referred to as the input-output table. This table exhibits diverse patterns depending on the different economic entities and statistical objects involved.

Timmer et al. [15] establish the World Input-Output Database (WIOD), an annual time-series database that captures transactions among 41 economic entities across 35 sectors, which is widely utilized in the literature. McNerney et al. [16] utilize WIOD to study the mechanisms by which production networks can amplify the effects of technological advancements and their subsequent impacts on long-term economic growth. Esfahani et al. [17] develop a novel method to quantify the driving forces behind global TFP and conduct a cross-country comparison based on WIOD. Trigg et al. [18] investigate the structure of Brazilian export linkages within global value chains using this database, arguing that the Brazilian economy heavily relies on primary-based industries. Del Río-Chanona et al. [19] utilize WIOD to analyze global trade from a complex network perspective, employing specific statistical indicators to measure the interdependence of input-output trade across the global market. Fagiolo et al. [20] utilize the database to construct indicators that explain the spatiotemporal distribution of per capita income across various countries from a network science perspective.

Many scholars have applied the input-output analysis framework to investigate sectoral linkages and their economic implications. Cella [21] disaggregates the total sectoral linkages into forward and backward linkages and compute these metrics. Clements [22] find there is a confusion inCella's disaggregation process to some extent, so he refines the measurement, enhance its economic interpretability. In addition to these, other researchers have employed alternative methods to measure sectoral interdependencies in an economy. For instance, Sonis [23] introduces the concept of a "field of influence', while Dietzenbacher [24] introduces the concept of the eigenvector method. These contributions make the combination of general equilibrium framework and input-output analysis possible to detect the function of sectoral linkages during the structural change process.

Relating to the structural change and development in China, several studies have examined various factors that influence this process. Researchers have identified several key elements as significant contributors to this transformation, including imperfect factor markets [25,26], the impact of international trade [27–29], increased investment in the services sector [30], the growing contributions of producer services and high-technology exporting manufacturing sectors [31], labor market dynamics [32], and the prevailing market structure dominated by state-owned enterprises [33].

In addition, many scholars utilize methods from input-output analysis to investigate structural change in the Chinese economy. For instance, Pan et al. [34] analyze the inter-industry technology spillover effects on labor productivity in the industrial sector. Their findings reveal that this effect has a significant positive impact, particularly pronounced among similar industries. As the Chinese economy develops and undergoes structural change, these inter-industry technology spillover effects have become increasing significant. Conversely, Cheng and Daniels [35] adopt a more focused approach, examining the roles and impacts of producer services in China. They point out five stylized facts regarding the development of this sector: the overall input ratio of services remains low; a majority of intermediate inputs are concentrated in traditional labor-intensive sectors; manufacturing emerges as the largest consumer of services, with intra-sector usage ranking second; research and development (R&D) activities are more significantly influenced by consumer services; and China demonstrates relatively low service input ratios across nearly all sectors.

In summary, current literature emphasizes the significance of sectoral linkages within the general equilibrium framework and through input-output analysis. Nevertheless, there is a relative scarcity of studies that specifically investigate sectoral linkages and their implications within the context of China. The mechanisms by which sectoral linkages operate during the production process, as determined by firms, require further investigation, as well as their influence on household consumption and demands for intermediate inputs within production networks. Traditional research on production networks has predominantly focused on shock propagation and economic fluctuations, with limited attention given to their roles in economic growth and associated structural change. The integration of the general equilibrium framework with input-output analysis necessitates additional exploration, particularly concerning microeconomic mechanisms. To address these gaps, our research aims to investigate the dynamics of structural change in China, focusing on the role of sectoral linkages throughout this process from both theoretical and empirical viewpoints. We incorporate production network analysis and input-output analysis into the general equilibrium framework to clarify the processes through which sectoral interconnections contribute to structural change in the Chinese economy. We aspire for our research to provide a novel contribution to the existing body of literature.

## Methodology

### Analytical framework

Drawing upon the existing body of literature and recognized stylized facts, this study formulates an economic model to investigate the impact of sectoral linkages on structural change within the Chinese economy.

The model consists of two types of agents: household and firms. In the goods market, there are three distinct types of final goods derived from the agricultural, manufacturing, and services sectors, while the factor markets are characterized by a single type of factor, labor. All markets are perfectly competitive. A representative household supplies labor to firms in three sectors, which compensate the household with wages in the factor market. The output produced by firms in three sectors can be either consumed by the household or utilized as intermediate goods input in production within the goods market. The intermediate goods input by firms across different sectors establish a production network within this economic framework, thereby inducing the significance of sectoral linkages and the underlying mechanisms that affect structural change within the economic system. This analytical framework is depicted in Fig 3 below.

The potential pathways through which sectoral linkages impact the structural change process can be discerned from Fig 3. Variations in the input of sectoral intermediate goods will influence the production processes of firms, subsequently leading to adjustment in their output levels. These changes in output will, in turn, affect the supply to satisfy household's demand as well as the demand from firms for intermediate inputs. Ultimately, all of these variations will propagate throughout the structural change process. The aforementioned analyses establish a connection between production networks and structural change.

The right-hand side of Fig 3 illustrates the connection of structural decomposition analysis (SDA) with the economic system. The labor factor is identified as the primitive input. The output employed by firms as intermediate goods in the production process is identified as the intermediate input. Additionally, the output consumed by household is identified as

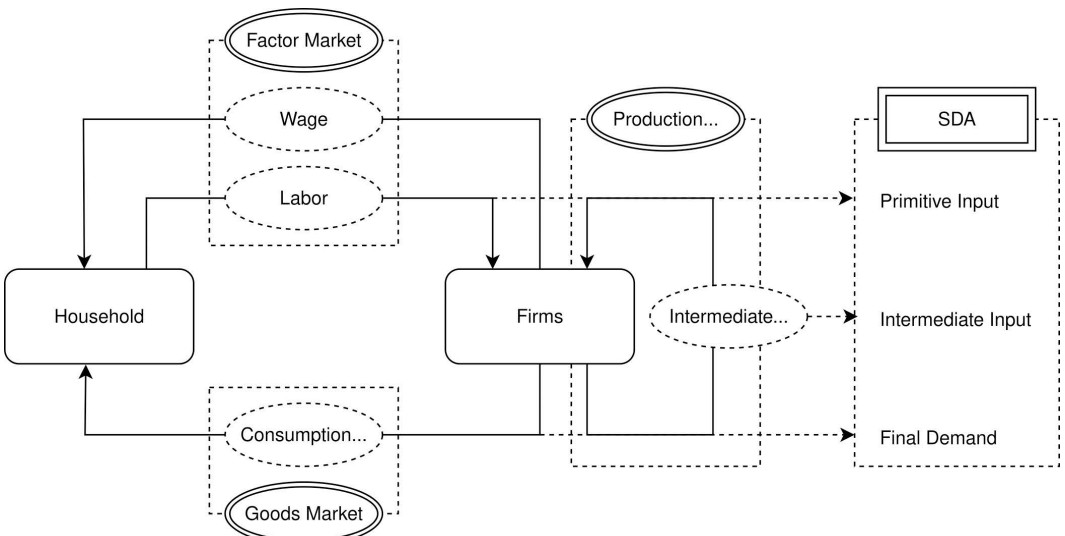

**Fig 3. Analytical framework.** The figure illustrates the fundamental structure of the economic system, along with its connection with the input-output analysis through the application of structural decomposition analysis (SDA).

final demand. Collectively, these three components constitute the primary driving forces within SDA. The SDA procedure part outlines the specific methodology of SDA and elucidates the roles of these three components.

Utilizing this analytical framework, we first develop a general equilibrium model related to the economic system depicted above. Subsequently, we calibrate the model parameters in accordance with the data, which is elaborated upon in the benchmark calibration part. Following the model specification and benchmark calibration, we conduct simulations using our baseline model and compare the results of these simulations with the observed data dynamics. To explore the roles of sectoral linkages and productivity as pivotal factors driving structural change in China, we perform a counterfactual analysis. Additionally, we adjust the preference and production function specification within our model to check its robustness. Drawing upon the results from these three procedures, we examine the underlying mechanisms from the perspective of production network. Finally, we integrate the model with SDA to identify and elucidate the roles of various components within the economic system. In conclusion, this section will encompass a discussion on model specification, benchmark calibration, counterfactual analysis and robustness check design, mechanism test design, structural decomposition analysis procedure literally.

## Model specification

As most literature argues and defines [37–39], structural change, or structural transformation, is often associated with the dynamics of three broad sectors. Informed by this body of literature, our model is constructed within the framework of a static, perfectly competitive open economy that encompasses three sectors: agriculture (denoted as $a$), manufacturing (denoted as $m$), and services (denoted as $s$).

**Firms.** Goods and labor are freely mobile between sectors. Within each sector, there exists a representative firm that employs labor and intermediate inputs to produce final goods. We adapt the Cobb-Douglas production function as referenced in [6,40]. The production function is as follows:

$$Y_i = Z_i(L_i)^{\alpha_i} \left[ \prod_{j \in \{a,m,s\}} (M_{ji})^{\mu_{ji}} \right]^{1-\alpha_i}, \quad \alpha_i, \ \mu_{jit} \in [0,1] \tag{1}$$

The term $M_{ji}$, $i$, $j \in \{a, m, s\}$ is the intermediate input from sector j that is utilized by sector $i$ in the production process. $L_i$ is the corresponding labor input. Exponential $\alpha_i$ is the share of nominal value-added relative to total output in sector $i$. Exponential $\mu_{ji}$ is the share of sector $j$'s goods in sector $i$'s total intermediate input, and $\sum_j \mu_{ji} = 1$. The term $Z_i$ represents the total factor productivity. All information presented in this section holds for a given period $t$. To avoid needless notation, for the remainder of this section I will not use the $t$ subscript.

**Household.** In the economy, there exists an infinitely living representative household endowed with $L$ units of labor per period. This household supplies labor inelastically to all firms.

At each period, the household purchases final goods from three sectors using its labor earnings. $C$ represents the aggregate consumption of final goods from these three sectors. In addition, the definition of consumption that we refer to here should be more precisely defined as final absorption. This includes consumption, investment, government purchases, and exports. For convenience, we denote it with the letter "C". The utility function is expressed in the Stone-Geary form, as described in [37]:

$$U(C) = \left[ \omega_a^{\frac{1}{\varepsilon}} \left( C_a + \overline{C_a} \right)^{\frac{\varepsilon-1}{\varepsilon}} + \omega_m^{\frac{1}{\varepsilon}} \left( C_m \right)^{\frac{\varepsilon-1}{\varepsilon}} + \omega_s^{\frac{1}{\varepsilon}} \left( C_s + \overline{C_s} \right)^{\frac{\varepsilon-1}{\varepsilon}} \right]^{\frac{\varepsilon}{\varepsilon-1}}$$

(2)

$\overline{C_a}$ is the level of subsistence consumption for agricultural final goods and $\overline{C_s}$ can be interpreted as the household's own production of services. These two variables cannot be zero simultaneously so as to keep preference non-homothetic.

The budget constraint is represented as follows: $P_a C_a + P_m C_m + P_s C_s = \sum_i WL_i = WL$, $i \in \{a, m, s\}$. $P_i$ is the price for the consumption of final goods in sector $i$. $W$ is the nominal wage in the labor market.

**Equilibrium.** In a competitive equilibrium, the following conditions are satisfied: (1) given the prices of goods, the household maximizes its utility under budget constraint; (2) given the prices of factors, firms maximize their profits under the production function; and (3) all markets for goods and factor clear.

(1) By solving the household's optimization problem, we obtain:

$$P = \left( \sum_i \omega_i P_i^{1-\varepsilon} \right)^{\frac{1}{1-\varepsilon}}$$

(3)

$$P_i \widetilde{C}_i = \omega_i \left( \frac{P_i}{P} \right)^{1-\varepsilon} \left( WL + \sum_i P_i \overline{C_i} \right)$$

(4)

$P$ is defined as the price of aggregate consumption $C$. $\widetilde{C}_i = C_i + \overline{C}_i$ for i=a, m, s, and $\overline{C_m} = 0$.

(2) By solving firms' optimization problems, we obtain:

$$WL_i = \alpha_i P_i Y_i$$

(5)

$$P_j M_{ji} = (1 - \alpha_i) \mu_{ji} P_i Y_i$$

(6)

So, the dual equation for (6) is:

$$P_i M_{ij} = (1 - \alpha_j) \mu_{ij} P_j Y_j$$

(7)

$M_{ij}$ is defined as the intermediate input from sector j that is utilized by sector $i$. According to the definitions of exponential $\alpha_i$ and $\alpha_j$, $WL_i$ and $WL_j$ can be regarded as the sectoral value-added; therefore, they can be denoted as $V_i$ and $V_j$. Consequently, $WL = W(\sum_i L_i) = \sum_i V_i = V$.

(3) The market clearing condition satisfies:

$$Y_i = C_i + \sum_j M_{ij}$$

(8)

$$\sum_i L_i = L$$

(9)

Using the equilibrium equations presented above, sectoral shares in value-added can be expressed as follows. More details can be found in Appendix A in S1 Appendix.

$$\frac{V_i}{V} = \left(1 + \frac{P_a \overline{C_a}}{V} + \frac{P_s \overline{C_s}}{V}\right) \left[\sum_j \Omega_{ij} \omega_j \left(\frac{P_j}{P}\right)^{1-\varepsilon}\right] - \Omega_{ia} \frac{P_a \overline{C_a}}{V} - \Omega_{is} \frac{P_s \overline{C_s}}{V}, \text{ for i = a, m, s.}$$

(10)

$\Omega_{ij}$, $\Omega_{ia}$ and $\Omega_{is}$ are corresponding entries in the matrix $\mathbf{\Omega}$, where $\mathbf{\Omega} = (\mathbf{I} - \mathbf{\Gamma})^{-1} \mathbf{A}$. Each entry in the matrix $\mathbf{\Gamma}$ is denoted as $\gamma_{ij} = \mu_{ij}(1 - \alpha_j)\left(\frac{\alpha_i}{\alpha_j}\right)$, for i, j=a, m, s. $\mathbf{A}$ is the diagonal matrix of $\alpha_i$. Equation (10) exhibits the formulation of sectoral value-added shares within a general equilibrium model, as well as their relationship to the sectoral linkages and prices.

**Benchmark calibration**

This subsection appliesdata to calibrate the aforementioned model to the Chinese economy for the period spanning from 1978 to 2014. Prior to the calibration process, we outline the data we utilize.

The data is collected from the World Input-Output Database (WIOD) [14]. For the years 1978–2000, the data is sourced from the Long-run WIOD [36], specifically the time series of the National Input-Output Tables (NIOT). For the subsequent years from 2001 to 2014, the data is sourced from the 2016 release version of WIOD. The subindustries are categorized into three broad sectors—agriculture, manufacturing, and services—according to the International Standard Industrial Classification of All Economic Activities, Rev. 3 (ISIC). The category of manufacturing encompasses not only itself but also includes other subindustries, as outlined in the following paragraph. For the sake of brevity, we collectively refer to these as manufacturing. This terminology is commonly used in relevant literature. The classification is widely used by researchers in the field of structural change [37,38].

In the Long-run WIOD, agriculture is classified under subindustries labeled by code AtB, manufacturing is classified under subindustries with codes C to F, and services is classified under codes G to LtQ. In the 2016 release version of WIOD (hereafter referred to as WIOD 2016), agriculture is classified under subindustries with codes A to B, manufacturing is classified under subindustries with codes C to F, and services is classified under subindustries with codes G to U. A notable distinction from conventional literature is our classification of construction within the manufacturing sector, which is consistent with the sector classification employed by the National Bureau of Statistics in China. Conversely, the International Standard Industrial Classification (ISIC) categorizes construction within the services sector.

Utilizing the classification mentioned above and data from WIOD, we construct the panel for calibration. The annual sectoral value-added, intermediate inputs and outputs are presented directly in the table. We aggregate them in accordance with the aforementioned classification. Since we assume that the output produced by firms is either utilized as sectoral intermediate inputs or consumed by the household as final goods, and all markets are perfectly competitive, so we aggregate the final absorption items in the table, which encompass consumption, investment and export, to match with our model.

The calculation of sectoral prices is conducted as follows. Given that WIOD provides sectoral final absorption data in both current and previous year prices, we can calculate the sectoral price deflator for each period. This is achieved by dividing the final absorption at current prices by the final absorption at the prices of the previous year. We designate 1978

as the base year, normalizing the price for each sector in that year to one. Subsequently, we derive the prices for subsequent years by dividing the growth factor of nominal final absorption by the growth factor of the price deflator.

We now proceed to the calibration procedure. The parameters $\alpha_{it}$ and $\mu_{jit}$ can be calculated directly from the data in the model specification. The parameters need to be calibrated are the preference parameters: $\omega_a$, $\omega_m$, $\omega_s$, and $\varepsilon$. We take the nonlinear least square (NLS) method [6] for calibrating these parameters. Using sectoral prices and total value-added, we select the manufacturing sector as a reference sector. The objective function for calibration aims to minimize the sum of squared distances between the actual sectoral final absorption, denoted as $E_{it}$, obtained from WIOD, and its corresponding values derived from the first-order conditions implied by Equations (4): $min_{\{\omega_i, \varepsilon\}} \sum_t \sum_{i \in \{a, s\}} [(\frac{P_{it}\overline{C_{it}}}{P_{mt}\overline{C_{mt}}}) - (\frac{E_{it}}{E_{mt}})]^2$. Through calculation and simplification, we get the final expression for the objective function, which is presented below. Additional information regarding the derivation of this equation can be found in Appendix B in S1 Appendix.

$$min_{\{\omega_i, \varepsilon\}} \sum_t \sum_{i \in \{a, s\}} \left[ \left( \frac{\omega_i}{\omega_m} \right) \left( \frac{P_{it}}{P_{mt}} \right)^{1-\varepsilon} - \frac{P_{it}\overline{C_i}}{\omega_m \left( V + P_{at}\overline{C_a} + P_{st}\overline{C_s} \right)} \left( \frac{P_t}{P_{mt}} \right)^{1-\varepsilon} - (\frac{E_{it}}{E_{mt}})\right]^2.$$

(11)

The calibration results are reported as follows in Table 1.

Sectoral productivity is defined as the residual in the production function, which can be calculated as follows:

$$Z_{it} = \frac{Y_{it}}{(L_{it})^{\alpha_{it}} \left[ \prod_{j \in \{a,m,s\}} (M_{jit})^{\mu_{jit}} \right]^{1-\alpha_{it}}}.$$

(12)

The nominal wage $W$ is derived from the expression: $W = \frac{V}{L}$. To accomplish this calculation, it's necessary to gather data on total labor input. We utilize data from the National Bureau of Statistics (NBS) and adjust it in accordance with the clarification proposed by Holz [41] to ensure consistency. It is noteworthy that the NBS of China revised its employment series data to correspond with the population censuses conducted in 2000 and 2010; however, data prior to 1990 remains unadjusted. Holz (2006) identified this discrepancy and addressed it by employing sectoral employment shares reported by the NBS, which were anchored to the population censuses of 1982 and 1990. As a result, this study integrates Holz's data for the years preceding 1990 with the NBS's data for the subsequent years.

## Counterfactual analysis and robustness check design

It can be deduced from Equation (10) that sectoral linkages and prices serve as the primary determinants influencing the dynamics of sectoral value-added shares. Furthermore, the forces that underlie sectoral prices are linked to sectoral productivities. Therefore, we employ the methodology presented in reference [29] for our counterfactual analysis to evaluate the impact of sectoral linkages and productivity. We analyze the former firstly. In this analysis, the parameters related to sectoral linkages $\Omega_{ij}$ are held constant at their initial values throughout all years, while all other parameters and variables remain unchanged. The differences observed between the results of this counterfactual analysis and the baseline will reflect the effects we seek to investigate.

**Table 1. Calibration results for parameters.**

| $\omega_a$ | $\omega_m$ | $\omega_s$ |
|---|---|---|
| 0.194 | 0.498 | 0.308 |
| $\overline{C_a}$ | $\overline{C_s}$ | $\varepsilon$ |
| -557.917 | 636.227 | 0.032 |

To examine the second potential channel through which these factors operate, it is essential to analyze the relationship between sectoral prices and their corresponding productivity, as articulated in Equation (13) of our basic model as below.

$$P_i = \frac{1}{Z_i} \left(\frac{W}{\alpha_i}\right)^{\alpha_i} \left\{ \prod_j \left[\frac{P_j}{(1-\alpha_i)\mu_{ji}}\right]^{\mu_{ji}} \right\}^{1-\alpha_i}.$$

(13)

This equation demonstrates that sectoral productivity has a significant influence on the equilibrium prices of final goods, which subsequently affects final absorption. We apply the same methodology previously discussed to analyze this relationship. In this case, sectoral productivity are held constant at its initial values from 1978 for all subsequent years, while other parameters and variables remain unchanged. The differences between the results obtained from this counterfactual analysis and the baseline will reflect the effects attributable to sectoral productivity.

In order to check the robustness of our model specification and the outcomes of the baseline simulation, we examine it from two dimensions: (1) alternative preference specification and (2) alternative production function specification.

Initially, we adopt a homothetic Constant Elasticity of Substitution (CES) utility function to characterize the preferences of a representative household, as exhibited in Equation (14).

$$U(C_t) = \left[\sum_i \omega_i^{\frac{1}{\varepsilon}} \left(C_{it}\right)^{\frac{\varepsilon-1}{\varepsilon}}\right]^{\frac{\epsilon}{\varepsilon-1}}.$$

(14)

The parameters and variable values are maintained in accordance with the baseline model. We simulate the dynamics and conduct counterfactual analyses utilizing the methodology previously outlined.

Subsequently, we replace the Cobb-Douglas production function with a generalized CES production function, as referenced in [42–44]. The functional form is articulated in Equation (15).

$$Y_i = Z_i[\alpha_i^{1-\rho}L_i^{\rho} + (1-\alpha_i)^{1-\rho}M_i^{\rho}]^{\frac{1}{\rho}}, \; M_i = [\sum_j \mu_{ji}^{\sigma}M_{ji}^{\sigma}]^{\frac{1}{\sigma}}.$$

(15)

For this production function, we apply the same methodology as previously described to derive the calibration functions for the two elasticity parameters, as well as the general equilibrium functions. We then simulate the dynamics of value-added shares and conduct the counterfactual analysis as previously indicated.

## Mechanism test design

In order to further elucidate the underlying mechanisms, this study refers to existing literature on production networks [7,9].

For the equilibrium equations previously articulated, we apply logarithmic transformation to both sides followed by total differentiation. This approach yields the subsequent equation (More details can be found in Appendix C S1 Appendix):

$$\boldsymbol{dlnY} \approx \boldsymbol{\Lambda_e dln\widetilde{C}} = \boldsymbol{\Gamma_z dZ} + \boldsymbol{\Gamma_p dlnRW}$$

(16)

In the Equation (16), $\boldsymbol{\Gamma_z} = \varepsilon\boldsymbol{\Lambda_e}\boldsymbol{\Lambda_{\mu}}\prime$ and $\boldsymbol{\Gamma_p} = (1-\varepsilon)\boldsymbol{\Lambda_e}\boldsymbol{A}$. Each diagonal entry of the matrix $\boldsymbol{\Lambda_e}$ is $\frac{2-\varepsilon}{\varepsilon}$ and each off-diagonal entry is $\frac{\varepsilon-1}{\varepsilon}$. $\boldsymbol{\Omega_{\mu}} = (\boldsymbol{I} - \boldsymbol{\Lambda_{\mu}})^{-1}\boldsymbol{A}$. Each entry of matrix $\boldsymbol{\Lambda_{\mu}}$ is denoted as $\lambda_{ij} = \mu_{ij}(1-\alpha_j)$, for i, j=a, m, s. The matrix $\boldsymbol{dlnRW}$ is the differentiation of market real wage, with all components expressed as $dlnW$-$dlnP$ uniformly.

Given that $Y_i = \alpha_i V_i$, it follows that $\boldsymbol{dlnY} = \boldsymbol{dlnV}$. Equation (16) exhibits the relationship between variations in total output or value-added and fluctuations in sectoral productivity and real wages. In a perfectly competitive factor market, there exists no sectoral heterogeneity regarding changes in real wages. Variations in sectoral productivity will propagate across

all sectors via sectoral linkages, as represented in matrix $\mathbf{\Gamma_z}$. According to reference [40], changes in sectoral productivity can be decomposed into intra-sectoral and inter-sectoral variations, denoted $dZ_i + \sum_j (\Omega_{\mu,ji} - \mathbf{1}_{j=i}) dZ_j$. $\mathbf{1}$ is the indicator function for j = i. The latter part illustrates how one sector may be influenced as a consumer of other sectors, which is characterized as the downstream network effect. In this context, sector linkages can act as a significant driving force.

In the mechanism test and SDA, we select the United States, Japan and South Korea as reference countries within the same time window. The United States has passed through the second phase of transformation and become service-oriented at the beginning of this period. The rapid growth of producer services, especially high R&D intensive services, has come to dominance the economy. Meanwhile, manufacturing production has largely been outsourced abroad to other countries. Japan has passed through the second phase of transformation in the mid-1970s, but its manufacturing sector also experienced rapid growth, which continues to play a substantial role in its economy until the economic bubble in the 1990s. It establishes tight connection with the services sector during this time. South Korea has completed its first phase of transformation during the earlier part of this period and passed through the second phase in the 1990s. In contrast with the United States and Japan, South Korea has relied on capital-intensive manufacturing sectors to complete its industrialization process and then developed services that are closely connected to support its manufacturing sector. These three countries have passed through the second phase of structural transformation process at varying points within the same period, resulting in diverse significance and developmental patterns within their manufacturing and services sectors. The distinct characteristics of structural change processes in these countries may offer valuable insights for the Chinese economy, particularly in terms of comparing the intersectoral linkages among different sectors. Additionally, all three countries—China, Japan, and South Korea—are situated within the East Asia region, where geographical proximity, cultural similarities, and comparable development patterns may yield analogous outcomes across these nations.

In order to compare the strength of linkages as a driving force across different sectors in China and three additional countries, we conduct a counterfactual analysis. In this analytical framework, we compare the downstream network effect coefficient $\sum_j (\Omega_{\mu,ji} - \mathbf{1}_{j=i})$ after assigning a value of 1 to all $dZ_i$.

## Structural decomposition analysis method

Structural Decomposition Analysis (SDA) is a widely used method employed to elucidate the channels through which each component operates within the input-output framework, applicable to both sectoral output and value-added metrics. This study applies SDA to identify the potential channels that drive structural change across three broad sectors.

From Equation (10), it indicates that $V_i = \sum_j \Omega_{ij} P_j \widetilde{C}_j$. This equation can be expressed in matrix form as follows:

$$\mathbf{V} = \begin{pmatrix} V_a \\ V_m \\ V_s \end{pmatrix}, \mathbf{L} = \begin{bmatrix} L_{aa} & L_{am} & L_{as} \\ L_{ma} & L_{mm} & L_{ms} \\ L_{sa} & L_{sm} & L_{ss} \end{bmatrix}, \hat{\mathbf{V}} = \begin{pmatrix} \alpha_a & 0 & 0 \\ 0 & \alpha_m & 0 \\ 0 & 0 & \alpha_s \end{pmatrix}, \mathbf{F} = \begin{pmatrix} P_a\widetilde{C_a} \\ P_m\widetilde{C_m} \\ P_s\widetilde{C_s} \end{pmatrix},$$

$$\mathbf{\Omega} = \begin{bmatrix} \Omega_{aa} & \Omega_{am} & \Omega_{as} \\ \Omega_{ma} & \Omega_{mm} & \Omega_{ms} \\ \Omega_{sa} & \Omega_{sm} & \Omega_{ss} \end{bmatrix}.$$

Therefore, $\mathbf{V} = \mathbf{\Omega F} = \mathbf{L}\hat{\mathbf{V}}\mathbf{F}$. To derive the matrix form for value-added shares, we divide the total value-added $\mathbf{V}$ across the three sectors. This can be articulated as: $\mathbf{v} = \mathbf{L}\hat{\mathbf{V}}\mathbf{f}$, where $\mathbf{v} = \begin{pmatrix} V_a/V \\ V_m/V \\ V_s/v \end{pmatrix}$ and $\mathbf{f} = \begin{pmatrix} P_a\widetilde{C_a}/V \\ P_m\widetilde{C_m}/V \\ P_s\widetilde{C_s}/V \end{pmatrix}$; $\mathbf{f}$ represents the total value-added driven by sectoral final absorption.

Within the SDA framework, the variations of $v$ can be decomposed into changes across its three components $L$, $\hat{V}$ and $f$. Given the discrepancies in the reference system, two distinct decomposition approaches are outlined below:

$$v_t - v_{t-1} = L_t \hat{V}_t f_t - L_{t-1} \hat{V}_{t-1} f_{t-1}$$

$$= (L_t - L_{t-1}) \hat{V}_{t-1} f_{t-1} + + L_t \left( \hat{V}_t - \hat{V}_{t-1} \right) f_{t-1} + L_t \hat{V}_t (f_t - f_{t-1})$$

$$= \Delta L \hat{V}_{t-1} f_{t-1} + L_t \Delta \hat{V} f_{t-1} + L_t \hat{V}_t \Delta f$$

An alternative approach can be expressed as:

$$v_t - v_{t-1}$$

$$= (L_t - L_{t-1}) \hat{V}_t f_t + L_{t-1} \left( \hat{V}_t - \hat{V}_{t-1} \right) f_t + L_{t-1} \hat{V}_{t-1} (f_t - f_{t-1})$$

$$= \Delta L \hat{V}_t f_t + L_{t-1} \Delta \hat{V} f_t + L_{t-1} \hat{V}_{t-1} \Delta f$$

These two separate decomposing approaches ultimately yield the same outcome, as evidenced in the existing literature. Dietzenbacher and Bart [45] investigate these two approaches and contend that employing the average of the results derived from both approaches is an acceptable strategy, which is frequently utilized in literature. Consequently, we adopt the averaging approach and reformulate it to derive the final decomposition expression.

$$v_t - v_{t-1}$$

$$= \frac{1}{2} \left( L_{t-1} \Delta \hat{V} f_t + L_t \Delta \hat{V} f_{t-1} \right) + \frac{1}{2} \Delta L \left( \hat{V}_{t-1} f_{t-1} + \hat{V}_t f_t \right) + \frac{1}{2} (L_{t-1} \hat{V}_{t-1} + L_t \hat{V}_t) \Delta f$$

The aforementioned expression suggests that changes in sectoral value-added shares can be ascribed to the changes in primitive inputs, the structure of intermediate inputs, and final demand, referred to as *primitive*, *intermediate* and *final* respectively. Corresponding with the model we specify aforementioned, these three components are matched with labor input and intermediate goods input in production by firms, final output consumed by the household. Therefore, we build the bridge between model framework and SDA method, so as to underline the microeconomic mechanism for SDA method.

## Quantitative results

### Baseline

Equation (10) demonstrates that both sectoral linkages and final absorption influence the value-added shares. In this study, we employ the aforementioned decomposition results to simulate the structural change occurring in China. The outcomes of this simulation are presented below and compared with the previously discussed data in Fig 4.

It can be found that the model successfully replicates the trend of structural change in China from 1978 to 2014. Fig 4 depicts a decline in the agricultural sector, a modest decline in manufacturing, and an increase in the services sector. This

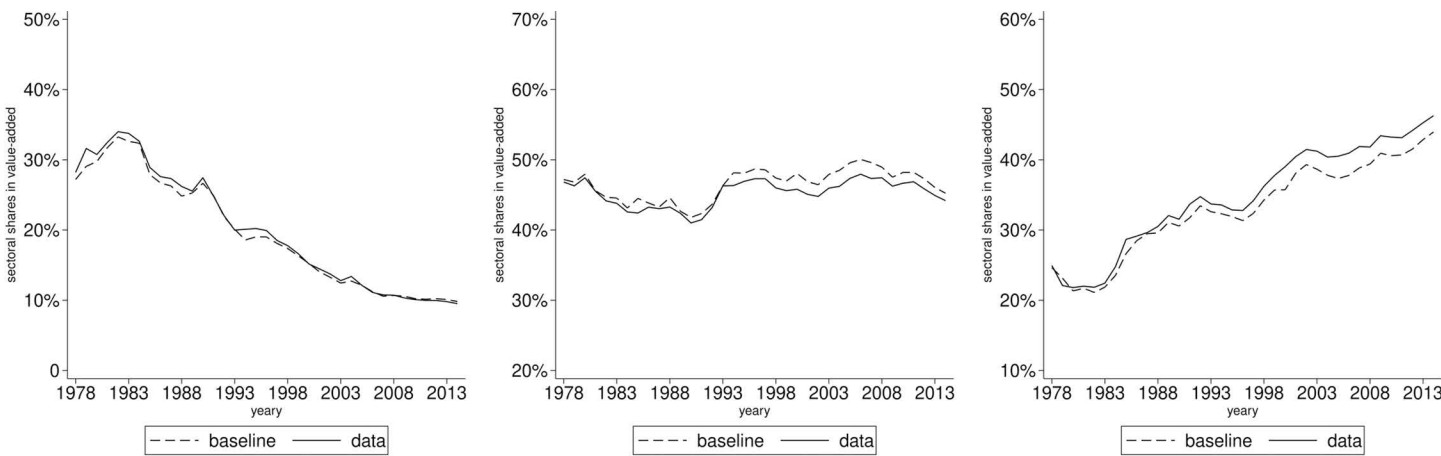

**Fig 4. Sectoral value-added shares: Baseline vs data.** The panels arranged from left to right illustrate the annual variations in sectoral value-added shares for agriculture, manufacturing and services. The dashed line depicts the model specified above, which is labeled as the baseline. The solid line depicts the actual data, which is calculated from WIOD.

observation is consistent with the data. The model demonstrates a close correspondence with the data across all three sectors, with discrepancies in value-added shares between the simulated results and the actual data remaining within a margin of 1.5 percentage points throughout the entire period. Specifically, compared to 1978, the value-added share of agriculture has diminished by 17.38 percentage points, while the actual data indicate a reduction of 18.69 percentage points. In the manufacturing sector, the share has decreased by 1.94 percentage points, whereas the data reflect a decrease of 2.65 percentage points. The services sector has seen an increase of 19.32 percentage points, compared to the data showing a rise of 21.33 percentage points. Consequently, the basic model effectively captures the characteristics of structural change in China. It also highlights the significant role of sectoral interconnections in production and other contributing factors in this transformation process.

## Counterfactual analysis

The basic model effectively captures the dynamics of sectoral shares in value-added, allowing us to identify the effects of input-output linkages and final absorption. We utilize the aforementioned methodology to conduct the counterfactual analysis. First, we will examine the former. Fig 5 presents the results of this analysis.

The findings indicate that the sectoral shares in value-added exhibit only minor fluctuation when sectoral linkages are maintained at their initial levels, a trend that is less pronounced than that observed in the baseline scenario. Specifically, Fig 5 illustrates a slight reduction in the share of agriculture, while the shares of manufacturing and services remain relatively stable. The changes in value-added shares for agriculture, manufacturing, and services from 1978 to 2014 are recorded as −11.35, 7.91, and 3.45 percentage points, respectively, compared to −17.38, −1.94, and 19.32 in the baseline. This suggests that sectoral linkages contribute to a 34.68% (6.03/17.38) decline in the agricultural share and an 82.15% (15.87/19.32) increase in the services sector. Notably, the manufacturing sector displays a slight upward trend, diverging from the decline indicated in the baseline scenario. This distinction is critical for comprehending structural changes within this sector.

In summary, sectoral linkages exert a significant influence on structural change arising from the intermediate absorption of production process across sectors. What cannot be overlooked is that the importance of input-output linkages for manufacturing development.

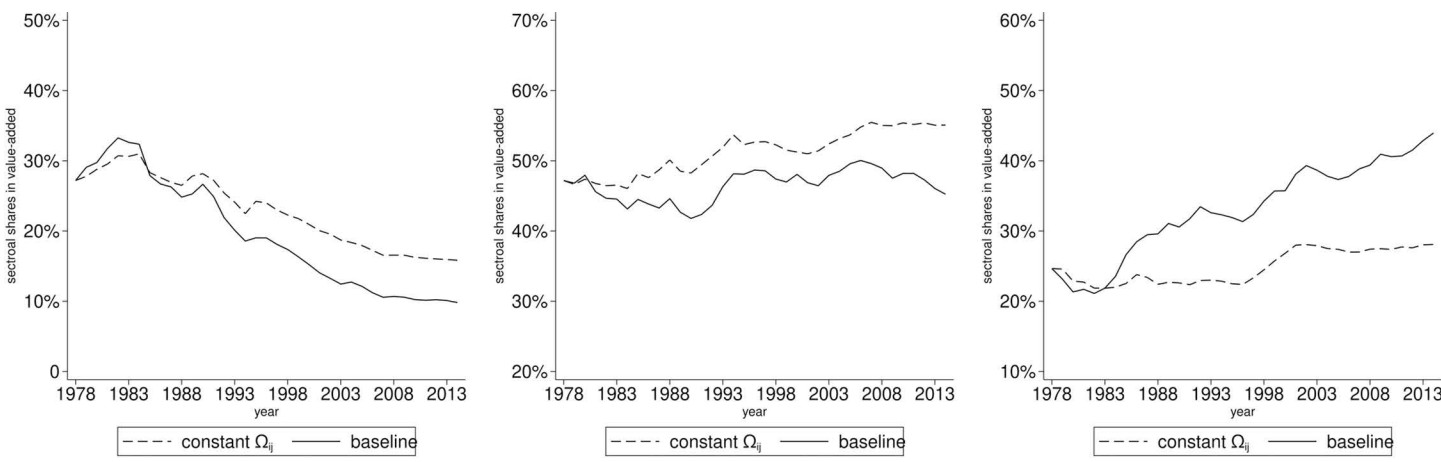

**Fig 5. Sectoral value-added shares: Constant linkages vs baseline.** The panels arranged from left to right illustrate the annual variations in sectoral value-added shares for agriculture, manufacturing and services. The dashed line represents a scenario in which the linkage parameters are set to their initial values, while all other parameters and variables remain constant. The solid line is depicted from the baseline model.

Next, we investigate the impact of final absorption on this issue. We employ the same methodology previously discussed. Fig 6 illustrates the results.

The counterfactual analysis reveals that sectoral shares in value-added vary differently when their productivity remain constant. Specifically, there is a slight decline in the agricultural sector, a pronounced decline in the manufacturing sector, and an increase in the services sector, as depicted in Fig 6. The changes in value-added shares for these three sectors from 1978 to 2014 are −11.71, −15.34, and 27.27 percentage points respectively, compared to −17.38, −1.94 and 19.32 in the baseline. This evidence further supports the notion of Baumol's cost disease affecting the manufacturing and services sectors, which can be attributed to their initial productivity gaps. This analysis indicates that, in the absence of technological advancements in sectoral productivity, the manufacturing share would experience a significant decline, while the services sector's share would see a substantial increase. In conclusion, sectoral productivity is crucial in affecting the final

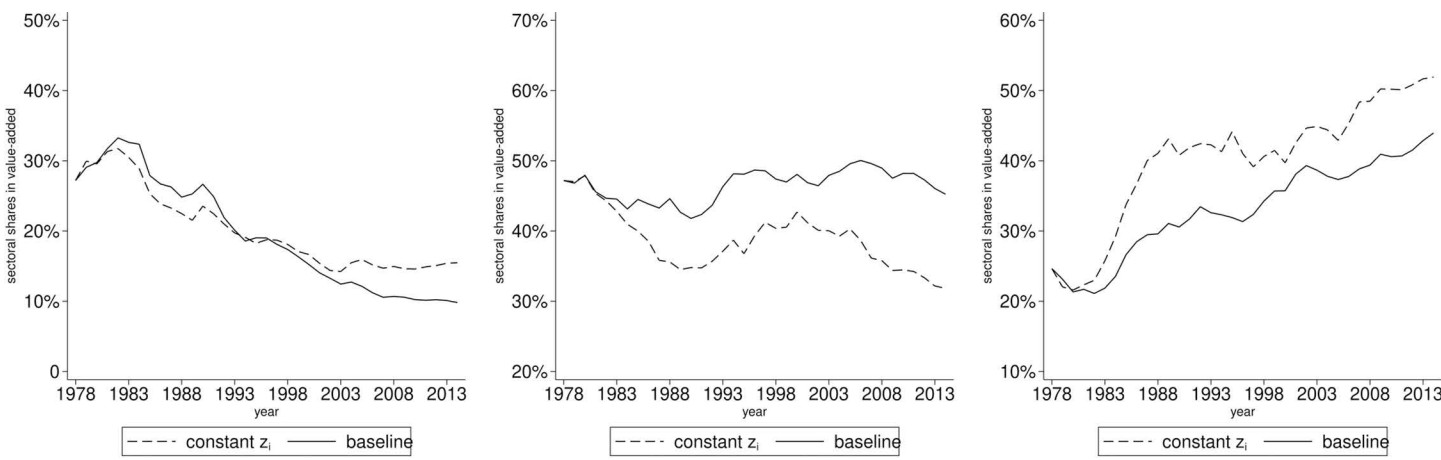

**Fig 6. Sectoral value-added shares: Constant productivity vs baseline.** The panels arranged from left to right illustrate the annual variations in sectoral value-added shares for agriculture, manufacturing and services. The dashed line represents a scenario in which sectoral productivity is set to their initial values, while all other parameters and variables remain constant. The solid line is depicted from the baseline model.

absorption through the pricing of final goods during the structural change, thereby highlighting their importance for industrial development.

## Robustness check

In this section, we employ the methodology mentioned above to check the robustness of our model specification results from two distinct dimensions: (1) different preference specification and (2) different production function specification.

Initially, we modify the preference specification and then simulate the dynamics and conduct the counterfactual analysis. The results from this preference specification indicate no obvious differences in the dynamics of sectoral value-added shares and the counterfactual outcomes when compared to the baseline model. Specifically, the changes in the three sector shares from 1978 to 2014 are recorded as −17.23, −2.07, and 19.31 percentage points from 1978 to 2014, compared to −17.38, −1.94, and 19.32 in the baseline. This suggests that sectoral linkages and productivity continue to be vital factors driving structural change. When the sectoral linkage parameters are set to their initial values, the observed changes in shares are −11.22, 7.79, and 3.44 percentage points, compared to −11.35, 7.91, and 3.45 in the baseline. Furthermore, when adjusting the sectoral productivity to its initial values, the changes in shares are −11.59, −15.4, and 27.21 percentage points, compared to −11.71, −15.34, and 27.27 in the baseline. Overall, the analysis reveals no significant differences between the two driving forces examined within this preference specification.

Subsequently, we replace the Cobb-Douglas production function with an alternative specification. The same methodology as described above is employed to derive the calibration function for the two elasticity parameters and corresponding general equilibrium functions. We then simulate the dynamics of value-added shares and conduct the counterfactual analysis as previously described. Under this new production function specification, the simulation results indicate that the value-added shares of the three sectors have changed by −18.67, −5.43, and 12.32 percentage points from 1978 to 2014, compared to −17.38, −1.94, and 19.32 in the baseline. Additionally, when the sectoral linkage parameters are set to their initial values, the changes in shares are reported as −11.15, 8.5, and 4.78 percentage points, in contrast to −11.35, 7.91, and 3.45 in the baseline. Moreover, when the sectoral productivity is adjusted to its initial values, the changes in sector shares are −12.73, −18.06, and 22.22 percentage points, compared to −11.71, −15.34, and 27.27 in the baseline. The specification of the CES production function similarly yields no significant differences between the two driving forces.

In summary, we conduct robustness checks by altering the preference and production specifications. There are no significant differences between these specifications and the baseline model in terms of simulation and counterfactual analysis. This finding reinforces our assertion about the functions of sectoral linkages and productivity in the structural change.

## Mechanism test

The counterfactual analysis and robustness checks presented above substantiate the significance of sectoral linkages. To further investigate the underlying mechanism, we take the methodology employed from the mechanism test design. The results are depicted in Fig 7 below.

Fig 7 demonstrates that sectoral linkages can bring in sectoral value-added growth through downstream network effect, given equivalent sectoral productivity growth. This phenomenon is pronounced in the agricultural sector within China and South Korea, where agricultural development increasingly relies on productivity growth in the other two sectors. In contrast, the manufacturing sector does not show an evident upward or downward trend and remains stable across these four countries. However, it exhibits a higher absolute value in China, indicating that the Chinese manufacturing sector constructs stronger interconnections with the other two sectors. The services sector displays varied dynamic patterns across these four countries. Notably, the Chinese services sector demonstrates the most substantial downstream network effect. This suggests that productivity growth in the other two sectors, particularly in manufacturing, can propagate downward to the services sector, with this effect being magnified through the established sectoral linkages.

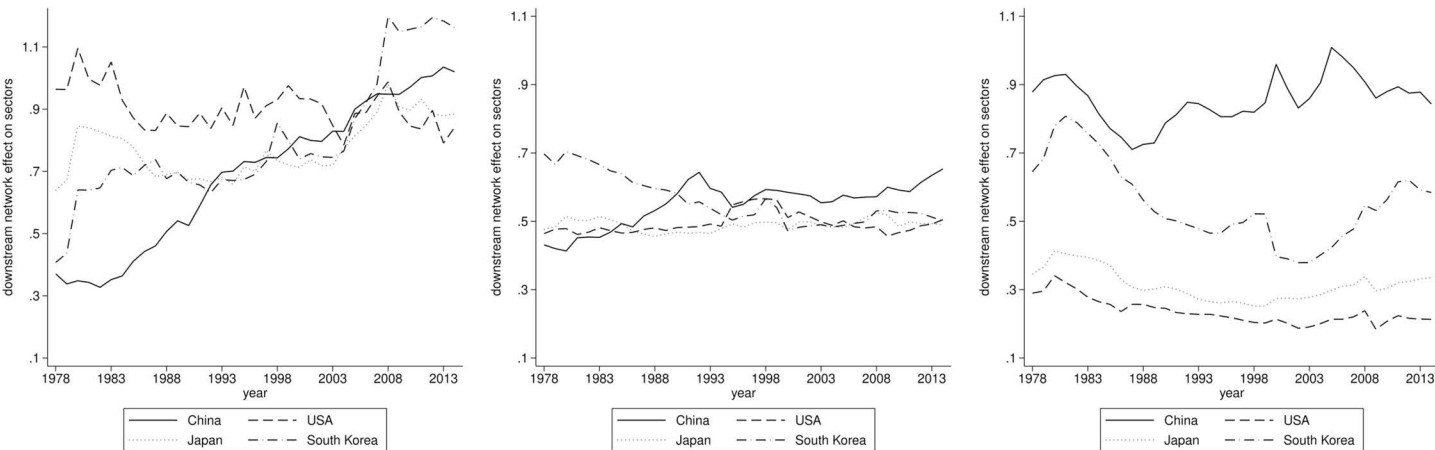

**Fig 7. The Downstream network effect on three sectors.** The panels arranged from left to right illustrate the annual variations of downstream network effect on agriculture, manufacturing and services for four countries mentioned above.

In summary, we examine the underlying mechanism of sectoral linkages, represented as downstream network effect, in the dynamics of sectoral value-added shares. The decomposition of sectoral value-added is made within a general equilibrium framework utilizing mathematical tools, followed by a counterfactual analysis. Our findings highlight considerable sectoral heterogeneities through cross-country comparisons. Specifically, in the context of China, all three sectors demonstrate stronger interconnections, whereby sectoral productivity growth exerts a more pronounced amplifying effect through these connections.

## Detecting driving forces via SDA

In this section, we analyze the function of different components of sectoral value-added shares through SDA method, which is based on input-output analysis. The decomposition approach previously described is applied to the sample data, allowing for an evaluation of the contributions of three distinct factors. The findings are summarized in Table 2.

Firstly, this study examines the individual impacts of these three factors. In comparison to the beginning of Chinese economic reforms in 1978, the contribution of primitive inputs has markedly decreased, particularly in comparison to three other countries. This decline reflects, to a certain extent, the industrial upgrading across three sectors. The reallocation of resources along both sides of the product chain at a micro-level induces the macro-level downward trend of primitive inputs. Meanwhile, sectoral linkages, represented by intermediate inputs, are increasingly critical. This finding supports our previous assertions of sectoral linkages, particularly within the manufacturing and services sectors. Furthermore, although final demand remains relatively stable, its significance should not be overlooked throughout this process.

Lastly, this study examines the impacts of these three factors within the sector. In the agricultural sector, the changes of these three factors occur simultaneously with its value-added share. This phenomenon is observable in both the Japanese and South Korean agricultural sectors. It is apparent that the final demand exerts the greatest influence, followed by the primitive inputs. This finding is closely associated with the production characteristics inherent to this sector. Agricultural production begins with the input of various resources to organize the production process, and the products are primarily sold directly to households and enterprises, both within and outside the sector. Consequently, there are relatively few interconnections with other sectors, which may explain why the intermediate inputs contribute a minimal amount.

In contrast, the manufacturing sector has witnessed a significant reduction in primitive inputs since 1978, as evidenced by the data presented in the table. During this period of development in China, it is widely recognized that Chinese

**Table 2. Factor contributions for structural change in China, The United States, Japan and South Korea[a].**

| Sectors | China | The United States | Japan | South Korea |
|---|---|---|---|---|
| **Agriculture** | −0.051 | 0.0007 | −0.0014 | −0.0018 |
| | −0.018 | −0.0058 | −0.0191 | −0.1491 |
| | −0.1187 | −0.0077 | −0.0117 | −0.0506 |
| | (−0.1877) | (−0.0128) | (−0.0322) | (−0.2015) |
| **Manufacturing** | −0.1855 | 0.0129 | −0.0115 | 0.0203 |
| | 0.1044 | −0.0275 | 0.009 | 0.0708 |
| | 0.0598 | −0.094 | −0.1249 | −0.041 |
| | (−0.0213) | (−0.1086) | (−0.1274) | (0.0501) |
| **Services** | −0.0334 | −0.0601 | −0.0126 | 0.0233 |
| | 0.1835 | 0.0798 | 0.0356 | 0.0364 |
| | 0.0589 | 0.1016 | 0.1367 | 0.0916 |
| | (0.209) | (0.1213) | (0.1597) | (0.1513) |

[a]The value in each blank for the 2rd to 5th columns reflects the contributions of three factors (primitive inputs, intermediate inputs and final demand) over the period from 1978 to 2014. The value in the parentheses at the end of each blank denotes the changes in total sectoral value-added shares.

manufacturing sector has become increasingly integrated into global production networks. Enterprises have engaged more with one another throughout the industrial upgrading process, both within the sector and outside. As a result, there has been a substantial increase in intermediate inputs, which counters the trend in value-added share and enhances its importance within the production framework.

Similarly, South Korea has also undergone more pronounced structural change compared to the United States and Japan during the same period. The data indicates that the intermediate inputs function like China during its manufacturing expansion. The final demand also holds considerable importance. As China's GDP per capita rises alongside its economic growth, a growing number of households are able to access a wider variety of products from the manufacturing sector. This trend is a natural consequence of the demand-driven effect associated with structural change. In contrast, the final demand exhibits a declining trend in the other three countries. Given their transition towards more service-oriented economies, the importance of manufacturing has become somewhat less significant.

The services sector is heavily dependent on sectoral linkages, as demonstrated by the dynamics of intermediate inputs. It is well established that services production require the exchange of substantial goods and information both within the sector and externally, thereby relying on sectoral linkages. Additionally, certain subindustries of this sector are directly connected to households, which amplifies the impact of final demand. When comparing China to other countries, it becomes apparent that intermediate inputs are more important in the growth of Chinese services sector, whereas the influence of final demand is more pronounced in the other three countries. This observation suggests distinct patterns of sectoral development. The growth of services in China demonstrates a high degree of reliance on sectoral linkages, while the other three countries exhibit a stronger connection to final demand.

The results of the structural decomposition analysis are consistent with our findings in the baseline. The intermediate inputs across three broad sectors exhibit similar patterns to their value-added shares within an economy throughout different stages of the structural change process. The growth and expansion of China's manufacturing and services sectors increasingly rely on sectoral linkages, thereby enhance their workforce in China's structural change process.

## Conclusion

The dynamic variations in sectoral linkages among three broad sectors that influence structural change are observed in many countries, including China. Our research develops a standard multisector model that incorporates sectoral

intermediate inputs into the production function to analyze the operational mechanisms of these linkages. Following this, we calibrate the model in accordance with Chinese economy, conduct counterfactual analysis, and check the robustness.

According to the dynamic variations of sectoral linkages, intermediate inputs are essential to the structural change process across three broad sectors. The impacts on a sector's value-added share can be attributed to both the final output inputs from other sectors and inter-sectoral inputs within the production process, particularly in the manufacturing and services sectors. The results of the counterfactual analysis suggest that without these variations, the share of manufacturing sector is likely to decline further, while the growth of the services sector will be considerably slower. Therefore, sectoral linkages can serve as a channel for the stability of manufacturing and the expansion of services during the structural change process.

Moreover, the traditional driving force of relative price changes resulting from shifts in sectoral productivity remains correlated through the final demand channel. Consistent with existing literature, it's observed that manufacturing share will experience a more pronounced decline, while the services sector will expand in the absence of productivity growth. This observation aligns with Baumol's concept of 'cost disease' from an alternative perspective.

To investigate the channel through which sectoral linkages affect value-added, our study explores the mechanism associated with production network research. Under conditions of equivalent sectoral productivity growth, we examine the downstream network effect across three sectors in China and three other countries. The comparative analysis reveals that China exhibits the greatest value of this effect within the manufacturing and services sectors, with a rapidly growing effect in the agricultural sector. These findings indicate that sectoral linkages can substantially amplify productivity growth, enabling downstream sectors to benefit from this effect, which is further advantageous for their value-added growth. We present these findings within the context of structural change in China from the perspective of production network.

In the structural decomposition analysis, we disaggregate value-added shares into primitive inputs, intermediate inputs, and final demand to evaluate their capacity to explain the dynamics of value-added shares and structural change. Given their distinct characteristics, the intermediate inputs and final demand, account for a great proposition of variations in value-added shares within the manufacturing and services sectors. This analysis is beneficial for understanding China's economic growth and structural change, as it reflects industrial upgrading and participation in the global value chain in unique ways.

Recognizing that the dynamic variations of sectoral linkages differ across various countries' production process, they may function in distinct ways. Our research has enabled us to identify their impact on Chinese economy, demonstrating their role in structural change within the country. However, our analysis should be further expanded, potentially through a more precise sectoral division perspective.

## Supporting information

**S1 Appendix. The Appendix presents the mathematical derivations for Equation (10) (pertaining to general equilibrium), Equation (11) (relevant to the calibration process), and Equation (16) (associated with the mechanism test).**
(DOCX)

**S1 Data. The dataset provides data for the simulation process outlined in this study, as well as for the cross-country comparative analysis. All data herein are computed based on the World Input-Output Database (WIOD).**
(XLSX)

## Author contributions

**Conceptualization:** Tao Jin.

**Data curation:** Zhihao Li.

**Formal analysis:** Tao Jin.

**Methodology:** Zhihao Li.

**Project administration:** Tao Jin.

**Software:** Zhihao Li.

**Writing – original draft:** Zhihao Li.

**Writing – review & editing:** Tao Jin.

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
