## [Decision Letter · Decision Letter 0]

3 Jun 2025

Dear Dr. Li,

We look forward to receiving your revised manuscript.

Kind regards,

Wisarut Suwanprasert, Ph.D.

Academic Editor

PLOS ONE

Journal Req

uirements:

Reviewers' comments:

Reviewer's Responses to Questions

**Comments to the Author**

1. Is the manuscript technically sound, and do the data support the conclusions?

Reviewer #1: Yes

Reviewer #2: Yes

2. Has the statistical analysis been performed appropriately and rigorously?

Reviewer #1: Yes

Reviewer #2: Yes

3. Have the authors made all data underlying the findings in their manuscript fully available?

Reviewer #1: Yes

Reviewer #2: Yes

4. Is the manuscript presented in an intelligible fashion and written in standard English?

Reviewer #1: No

Reviewer #2: Yes

Reviewer #1: This manuscript investigates the role of sectoral input-output linkages in the structural transformation of the Chinese economy by employing an integrated approach that combines input-output analysis with a general equilibrium model. The authors address a relatively under-explored but increasingly significant perspective on long-term economic development.

While the topic is timely and the theoretical framework is promising, the current manuscript suffers from serious structural, conceptual, and presentation-related issues. These issues must be addressed comprehensively for the manuscript to achieve publishable quality. I suggest that this paper be resubmitted after a rigorous revision to amend flaws in structure and flow.

1. Introduction Lacks Clear Motivation and Context

The introduction needs clearer contextual framing of the research problem. The authors should explicitly state the time period, motivating data trends, and empirical relevance of studying structural transformation in China.

This section reads more like a summary rather than an introduction that explains the context and the necessity of the study, in light of both the literature gap and practical demand. It should clarify why sectoral linkages are essential beyond what existing studies have covered.

While the discussion of the literature gap is present, it is not sufficiently compelling.

2. Literature Review Is Unstructured

While the manuscript references several relevant studies, the literature review lacks thematic coherence. Although the authors mention three main research gaps (structural change, production networks, and input-output analysis), this structure is not reflected in the review section.

The writing should be more concise and should avoid repetitive or generic statements.

3. Modeling Section Lacks Analytical Clarity

The paper combines multiple methodologies but fails to provide a guiding framework that clarifies how these components fit together.

Moreover, the absence of a diagram or analytical framework makes it challenging to follow the model's logic and flow.

4. Results and Discussion Are Blended Without Clear Distinction

The manuscript merges methodology, empirical results, and interpretation into a single narrative, which reduces readability.

Important results regarding China are integrated into the data/methods section, while cross-country comparisons (with the USA, Japan, and South Korea) appear suddenly without introduction or justification.

I suggest clearly separating methods and results. The key findings on China’s sectoral transformation should be moved to the results section. Additionally, introduce and justify the inclusion of cross-country comparisons earlier in the manuscript.

Minor Issues

The Facts Section should be integrated into the Introduction Section, the current manuscript length should highly be reduced.

Reviewer #2: Areas for Improvement

Enhance the transparency of the calibration process by providing more details on: (1) assumptions of I-O model and paper's assumptions (perfect competition), especially your dataset (sector aggregation), eq 11.

clarify the reasons for the selection of comparator countries, why not depend on other indicators.

Consider submitting the dataset for review (you have linked the sources)

shorten the other contents of comparator countries, delve deep the insights of china sector linkages.

**Do you want your identity to be public for this peer review?** For information about this choice, including consent withdrawal, please see our Privacy Policy

Reviewer #1: No

Reviewer #2: No

---

## [Author Response · Author response to Decision Letter 1]

17 Jul 2025

Dear Editors and Reviewers,

We would like to sincerely thank you for your time and effort in reviewing our manuscript titled “Sectoral linkages and their influence on structural change: the case of China” (PONE-D-25-11213). We greatly appreciate your insightful and constructive comments and advice, and we have carefully addressed these concerns and made a proper revision of the manuscript. These comments and suggestions have not only enabled us to provide a highly improved manuscript but also inspired us to conduct more in-depth studies in future works. The main corrections in the paper and the responds to the reviewers’ comments are stated below.

Reviewer #1:

1. Introduction Lacks Clear Motivation and Context.

The introduction needs clearer contextual framing of the research problem. The authors should explicitly state the time period, motivating data trends, and empirical relevance of studying structural transformation in China.

This section reads more like a summary rather than an introduction that explains the context and the necessity of the study, in light of both the literature gap and practical demand. It should clarify why sectoral linkages are essential beyond what existing studies have covered.

While the discussion of the literature gap is present, it is not sufficiently compelling.

Response: We have revised the Introduction section and rearrange the structure. The introduction is restructured in three parts: the context and stylized facts about structural transformation process in Chinese economy; the motivation and research questions arisen from these observed facts; the contributions of our research and the structure of the study.

Firstly, we give the definition of structural transformation and provide relevant context regarding the development of the Chinese economy. We utilize data from the World Input-output database (WIOD) to depict the dynamics of value-added shares and sectoral linkages throughout the structural transformation process in China from 1978 to 2014.

The trends in value-added shares for the agricultural and services sectors exhibit common features which is argued in the so called ‘Kuznets Facts’: the descending trend of agriculture and ascending trend of services. In contrast, the manufacturing sector doesn’t display an evident hump-shaped path but is more stable and the share decreases a little in 2014 compared to 1978. These constitute the characteristics of structural transformation process within Chinese economy.

We use the shares of intra and inter sectoral intermediate inputs for three broad sectors to depict the dynamics of sectoral linkages. The share of agricultural intermediate inputs exhibits the descending trend in three sectors, which is similar to the dynamic of value-added share. What are the different is that the descending trend of intermediate input from services sector is only evident in the services intra-sector production. Meanwhile, the slightly descending trend of manufacturing sector is only evident in the services sector while it exhibits the opposite slightly ascending trend in agriculture and manufacturing sectors. Since the dynamics of sectoral intermediate inputs directly connect to the production process, which encompass the influence of sectoral linkages on sectoral output and the value-added, it puts forward our necessity to study the impacts of sectoral linkages on structural transformation in China economy.

Secondly, after we illustrate the stylized facts about the structural transformation process in China, we lay out the motivation to investigate our research theme: impacts of sectoral linkages on this process. How does the sectoral linkages work during this process and via which channels? How does the different structures of it in the three broad sectors contribute to the dynamics of sectoral growth? By echoing these questions, we attempt to examine the impacts of sectoral linkages from a long-term sectoral growth perspective.

Finally, we argue about our contribution from three perspectives: the driving forces of structural transformation; production networks and economic development; input-output analysis. We complement the existing literature by emphasizing that the utilization of intermediate goods in production and their changes are as essential as the demand for final goods and technological advancements in understanding sectoral value-added dynamics. We examine the functions of sectoral linkages from the production view and its connection with final output and consumption. Traditional literature provides a lot of evidence for the horizontal fragmentation among sectors have great impact on the structural transformation process from the demand-driven view. We provide the evidence for the function of vertical fragmentation from the intermediate inputs view. It highlights the long-term effects of productivity growth propagating through production networks. Our finding confirms that production networks not only affect aggregate volatility and economic fluctuations in the short term but also influence sectoral value-added dynamics in the long term. We make a connection of different components in sectoral value-added shares under the SDA framework with their counterparts in the general equilibrium model. It establishes a theoretical foundation and microeconomic mechanism for the use of SDA, which could enhance its application in research.

2. Literature Review Is Unstructured.

While the manuscript references several relevant studies, the literature review lacks thematic coherence. Although the authors mention three main research gaps (structural change, production networks, and input-output analysis), this structure is not reflected in the review section.

The writing should be more concise and should avoid repetitive or generic statements.

Response: We have revised the literature review and restructured it in accordance with three main gaps you mentioned above.

The first relevant body of literature examines the economic implications of changes in sectoral linkages through cross-country comparisons. We review the studies that analyze the impacts of these linkages on sectoral TFP and income levels. They argue that wealthier nations exhibit tighter and less distorted sectoral linkages compared to their poorer counterparts, thereby contributing to a reduction in income disparities. Then we review the studies regarding the servitization of sectoral intermediate inputs in cross-country comparisons. This phenomenon is particularly pronounced in wealthier countries, which results in a more dominant service sector in these economies.

The second relevant body of literature focuses on the role of sectoral linkages within production networks. Some studies argue about the function of these linkages in propagating economic shocks and inducing fluctuations. They examine both upstream and downstream effects of linkages propagating shocks in a production network. Other studies argue about the implications of these linkages in transferring distortions and misallocations in an inefficient economy, with a particular emphasis on productivity and income outcomes.

The third relevant body of literature investigates the effects of sectoral linkages through input-output analysis. We begin by reviewing the introduction and its subsequent development of this methodology, focusing on the implementation of WIOD database. Following this, we analyze literature relevant to this research theme within the input-output framework. They construct various indicators and methodologies to identify the extent of sectoral linkages based on the input-output table data, which make the combination of general equilibrium models with input-output analysis framework possible.

Finally, we review literature investigating the different factors contributing to the structural transformation process in Chinese economy. They mainly discuss about these aspects: the structure of factor markets, international trade, technology spillovers, and investment and development within the service sectors.

By reviewing the relevant literature, we point out their limitations and what we have done in our research. The mechanisms by which sectoral linkages operate during the production process, as determined by firms, require further investigation, as well as their influence on household consumption and demands for intermediate inputs within production networks. Traditional research on production networks has predominantly focused on shock propagation and economic fluctuations, with limited attention given to their roles in economic growth and associated structural changes. The integration of the general equilibrium framework with input-output analysis necessitates additional exploration, particularly concerning microeconomic mechanisms. To address these gaps, our research aims to investigate the dynamics of structural change in China, focusing on the role of sectoral linkages throughout this process from both theoretical and empirical viewpoints. We incorporate production network analysis and input-output analysis into the general equilibrium framework to clarify the processes through which sectoral interconnections contribute to structural changes in the Chinese economy.

3. Modeling Section Lacks Analytical Clarity.

The paper combines multiple methodologies but fails to provide a guiding framework that clarifies how these components fit together.

Moreover, the absence of a diagram or analytical framework makes it challenging to follow the model's logic and flow.

Response: In light of this and the next one comment, we have revised the structure of the paper and add the analytical framework subsection in the Methodology section.

This subsection presents the construction of an economy system and the hypotheses to specify our model. The economy system encompassed a goods market, a factor market and two distinct types of agents. The goods market comprises three categories of final goods from the agricultural, manufacturing and services sectors. The factor market is characterized by a single production factor, labor. All the markets are perfectly competitive. Two types of agents include a representative household and various firms across the three sectors. The household provides labor to the firms, which in turn compensate the household with wages in the factor market. The output produced by the firms can either be consumed by the household as final demand or utilized by the firms as intermediate goods in the production process within the goods market.

The intermediate goods input by firms across different sectors establish a production network within this economic framework, thereby inducing the significance of sectoral linkages and the underlying mechanisms that affect structural transformation within the economic system. Variations in the input of sectoral intermediate goods will influence the production processes of firms, subsequently leading to adjustments in their output levels. These output changes will subsequently affect the supply to meet household demand as well as the demand from firms for intermediate inputs. Ultimately, all of these variations will propagate throughout the structural transformation process. The analyses presented establish a connection between production networks and structural transformation.

The right side of the figure illustrates the connection of structural decomposition analysis (SDA) and the economic system. The labor factor is recognized as the primitive inputs, while the output utilized by firms as intermediate goods in the production process is recognized as the intermediate inputs. Furthermore, the output consumed by the household is recognized as final demand. Collectively, these three components represent the primary driving forces within the SDA. The SDA Procedure subsection outlines the specific methodology of SDA and clarifies the roles of these three components.

By constructing the economic system and establishing connections among various research methodologies, we propose the analytical framework for our study.

4. Results and Discussion Are Blended Without Clear Distinction.

The manuscript merges methodology, empirical results, and interpretation into a single narrative, which reduces readability.

Important results regarding China are integrated into the data/methods section, while cross-country comparisons (with the USA, Japan, and South Korea) appear suddenly without introduction or justification.

I suggest clearly separating methods and results. The key findings on China’s sectoral transformation should be moved to the results section. Additionally, introduce and justify the inclusion of cross-country comparisons earlier in the manuscript.

Response: We adjust the structure and separate these contents into the Methodology Section and Quantitative Results Section. The Methodology Section exclusively encompasses comprehensive procedures, which are systematically arranged into the following subsections: model specification, calibration, counterfactual analysis and robustness check design, mechanism test design, and the SDA method. All the relevant results are compiled within the Quantitative Results.

The Methodology Section starts at outlining the analytical framework necessary for formulating our research objectives and the connection among various analysis patterns, as detailed previously. Then we specify our model structure: the preference of the representative household and the production function that incorporates sectoral linkages for firms. We provide a comprehensive description of the optimization procedure employed to achieve general equilibrium within the Model Specification subsection. The expression of sectoral value-added shares for calibration and subsequent analysis is presented at the end of this subsection. This expression highlights two primary factors influencing structural change: sectoral linkages and productivities. In the Benchmark Calibration subsection, we discuss about the data from WIOD in our research and outline the classification for categorizing the raw data into three broad sectors. The calibration procedure for model parameters, along with the adjustments made to variables, is subsequently detailed, followed by the outcomes of the calibration.

In the counterfactual analysis design, we examine the function of two aforementioned primary factors. We organize this analysis by setting the variables of interest to their primitive values while maintaining all other parameters and variables constant. The differences between the results of this counterfactual analysis and the baseline outcomes will illuminate the effects we seek to investigate. In the robustness check design, we sequentially modify the utility function for the household and the production function for firms one after another. Specifically, we alter these two functions into the constant elasticity of substitution (CES) form and conduct the simulations and counterfactual analyses. We then we compare the results from this procedure with the baseline to check our model’s robustness.

The mechanism test design exhibits the relationship between variations in sectoral value-added and the variations in sectoral linkages and productivities in details. This connection incorporates network effects and corresponding analyses. The final subsection demonstrates how to integrate the SDA method from input-output analysis with the general equilibrium model. We provide detailed procedures for decomposing sectoral value-added shares into three components: primitive inputs, intermediate inputs and final demand, as part of the SDA. These components are matched with labor input and intermediate goods input in production by firms, final output consumed by the household in our model. Consequently, we establish a connection between the model framework and the SDA method, thereby emphasizing the microeconomic mechanisms underlying the SDA approach.

The Quantitative Results Section presents the findings derived from the methodology outlined in the Methodology Section. Based on the expression of sectoral value-added shares and the calibration of parameters, we simulate their dynamics and compare the results with the actual data. The gaps observed across three broad sectors are relatively small, indicat

---

## [Editor Report · Decision Letter 1]

8 Aug 2025

Sectoral linkages and their influence on structural change: the case of China

PONE-D-25-11213R1

Dear Dr. Li,

We’re pleased to inform you that your manuscript has been judged scientifically suitable for publication and will be formally accepted for publication once it meets all outstanding technical requirements.

Kind regards,

Wisarut Suwanprasert, Ph.D.

Academic Editor

PLOS ONE
---

## [Editor Report · Acceptance letter]

PONE-D-25-11213R1

PLOS ONE

Dear Dr. Li,

I'm pleased to inform you that your manuscript has been deemed suitable for publication in PLOS ONE. Congratulations! Your manuscript is now being handed over to our production team.

Kind regards,

on behalf of

Dr. Wisarut Suwanprasert

Academic Editor

PLOS ONE